

# A global compilation of in situ aquatic high spectral resolution inherent and apparent optical property data for remote sensing applications

Kimberly A. Casey[1,2], Cecile S. Rousseaux[1,3,4], Watson W. Gregg[1,3] Emmanuel Boss[5],
Alison P. Chase[5], Susanne E. Craig[4,6], Colleen B. Mouw[7], Rick A. Reynolds[8], Dariusz Stramski[8],
Steven G. Ackleson[9], Annick Bricaud[10], Blake Schaeffer[11], Marlon R. Lewis[12], Stéphane
Maritorena[13]

[1]Earth Sciences Division, NASA Goddard Space Flight Center, Greenbelt, MD, 20771, USA
[2]Land Resources, U.S. Geological Survey, Reston, VA, 20192, USA
[3]Global Modeling and Assimilation Office, NASA Goddard Space Flight Center, Greenbelt, MD, 20771, USA
[4]Universities Space Research Association, Columbia, MD, 20771, USA
[5]School of Marine Sciences, University of Maine, Orono, ME, 04469, USA
[6]Ocean Ecology Laboratory, NASA Goddard Space Flight Center, Greenbelt, MD 20771, USA
[7]University of Rhode Island, Graduate School of Oceanography, Narragansett, RI, 02882, USA
[8]Marine Physical Laboratory, Scripps Institution of Oceanography, University of California San Diego, La Jolla,
CA, 92093, USA
[9]Naval Research Laboratory, Washington, D.C., USA, 20375
[10]CNRS and Sorbonne Université, Laboratoire d'Océanographie de Villefranche (LOV), F-06230 Villefranche-sur-
mer, France
[11]Office of Research and Development, U.S. Environmental Protection Agency, USA
[12]Department of Oceanography, Dalhousie University, Halifax, Nova Scotia, Canada
[13]Earth Research Institute, University of California, Santa Barbara, CA, 93106, USA

*Correspondence to*: Kimberly A. Casey (Kimberly.A.Casey@nasa.gov)

**Abstract.** Light emerging from natural water bodies and measured by remote sensing radiometers contains
information about the local type and concentrations of phytoplankton, non-algal particles and colored dissolved
organic matter in the underlying waters.  An increase in spectral resolution in forthcoming satellite and airborne
remote sensing missions is expected to lead to new or improved capabilities to characterize aquatic ecosystems.
Such upcoming missions include NASA's Plankton, Aerosol, Cloud, ocean Ecosystem (PACE) Mission; the NASA
Surface Biology and Geology observable mission; and NASA Airborne Visible / Infrared Imaging Spectrometer -
Next Generation (AVIRIS-NG) airborne missions. In anticipation of these missions, we present an organized dataset
of geographically diverse, quality-controlled, high spectral resolution inherent and apparent optical property
(IOP/AOP) aquatic data.  The data are intended to be of use to increase our understanding of aquatic optical
properties, to develop aquatic remote sensing data product algorithms, and to perform calibration and validation
activities for forthcoming aquatic-focused imaging spectrometry missions.  The dataset is comprised of
contributions from several investigators and investigating teams collected over a range of geographic areas and
water types, including inland waters, estuaries and oceans.  Specific in situ measurements include coefficients
describing particulate absorption, particulate attenuation, non-algal particulate absorption, colored dissolved organic
matter absorption, phytoplankton absorption, total absorption, total attenuation, particulate backscattering, and total



backscattering, as well as remote-sensing reflectance, and irradiance reflectance. The dataset can be downloaded

from https://doi.pangaea.de/10.1594/PANGAEA.902230 (Casey et al., 2019).

**1 Introduction**

Remote sensing of Earth's aquatic areas is a powerful tool for understanding water quality, aquatic and ecological dynamics, and the concentrations and types of phytoplankton, colored dissolved organic matter and non-algal particles present over time. Aquatic remote sensing initially focused on chlorophyll-a concentration ([Chl]) (NASA

GSFC, Ocean Biology Processing Group, 2014) which serves as a proxy for understanding the distribution of phytoplankton biomass. The most widely used approach to estimate [Chl] has been empirical relationships between band ratios or band differences of remotely sensed reflectance and [Chl] (O'Reilly et al. 1998; Hu et al. 2012). Chlorophyll-a concentration estimated from aquatic color has been studied for many decades and remote sensing retrievals are well validated (McClain, 2009). Chlorophyll algorithm improvements continue in response to

enhanced spectral resolution and sensor capabilities of upcoming Earth Observation missions (O'Reilly and Werdell, 2019). Aquatic remote sensing is now being further used to aid the understanding of more complex dynamics including atmosphere–ocean heat exchange and the role and feedback effects of aquatic constituents, as well as alteration of phytoplankton community structure in a changing climate (Kim et al., 2018; Dutkiewicz et al., 2019; Del Castillo et al., 2019). These analysis approaches involve numerical modeling and analyzing radiometric

variability of many spectral bands.

In situ data is a key requirement for aquatic remote sensing algorithm development, validation and calibration activities and for advancing our aquatic remote sensing data capabilities. The in situ data provided in this manuscript include inherent optical properties (IOPs) and apparent optical properties (AOPs) from a wide

distribution of aquatic environments and geographic locations. Briefly, inherent optical properties are the light absorption and scattering properties of the natural waters which are dependent solely on the concentrations and composition of water constituents irrespective of the illumination field within a water body. An apparent optical property is an optical property that can be used as a descriptor of a water body and is primarily dependent on the IOPs of the aquatic medium and, to a lesser degree, on the directional structure of the ambient radiance distribution

within a water body. In this article, we provide data for the AOPs of irradiance reflectance ($R$) and radiance-reflectance or remote-sensing reflectance ($R_{rs}$) just above the water surface, and for the IOPs representing absorption and backscattering coefficients of natural waters. The spectral IOPs (where $\lambda$ is light wavelength in a vacuum) can be partitioned into the absorption due to water itself ($a_w(\lambda)$, m⁻¹), phytoplankton ($a_{ph}(\lambda)$, m⁻¹), non-algal particles ($a_{nap}(\lambda)$, m⁻¹), colored dissolved organic matter ($a_{cdom}(\lambda)$, m⁻¹), and backscattering due to water itself ($b_{bw}(\lambda)$, m⁻¹)

and particles ($b_{bp}(\lambda)$, m⁻¹).

With coincident high spectral resolution in situ IOP and AOP data, scientists can better develop and validate aquatic remote sensing algorithms to derive IOPs from measured AOPs (e.g. Werdell et al., 2018, and references therein).



Torrecilla and others (2011) demonstrated that hyperspectral data of phytoplankton absorption and remote-sensing
reflectance provide improved discrimination of dominant phytoplankton groups in open-ocean environments
compared with multi-spectral data. High spectral resolution aquatic remote sensing significantly improves retrievals
of optical constituents in inland, coastal and polar aquatic environments, where these environments exhibit
significant smaller-scale temporal and spatial variability, increased decoupling between in-water constituents, and a
greater dynamic range in parameter values compared to the open ocean (Mouw et al., 2015; Bell et al., 2015;
Dierssen et al., 2015; Hu et al., 2015; Vandermeulen et al., 2017). In inland, coastal and polar aquatic areas,
dissolved organic matter (DOM) and non-algal particles (NAP) play a more important role in affecting the color of
water as well as its biogeochemistry, sediment transport, and primary productivity (Devred et al., 2013; Mouw et al.,
2017). Thus, greater measurement precision is desirable. Carbon pools are also varied in inland and coastal
environments due to riverine inputs, terrestrial influence, resuspension and mixing requiring greater spectral
resolution and broader spectral range to differentiate the spectral slope of CDOM sources. Further, there are
increased instances of harmful algal bloom formation in many aquatic environments. Some harmful algal blooms
can be discriminated based on their unique optical signatures and therefore additional spectral bands beyond the
current multi-spectral capabilities would be highly beneficial (Wang et al. 2016, Pahlevan et al., 2019). Moving
geographically to polar latitudes, Neukermans and others (2016) demonstrated improved discrimination of
planktonic communities in the Arctic by using hyperspectral instead of multispectral satellite data. In short, remote
sensing capabilities in all aquatic environments are expected to improve considerably in precision and accuracy with
high radiometric quality high spectral resolution measurements.

We summarize many of the historic, current and forthcoming high spectral resolution satellite missions potentially
applicable to aquatic remote sensing goals in Figs. 1 and 2. High spectral resolution technological demonstration
satellite missions that have flown or are currently in operation or late planning stages are detailed as follows. One of
the longest spaceborne hyperspectral data records is provided by NASA's EO-1 Hyperion sensor, which was
launched on 21 November 2000 and decommissioned on 22 February 2017. Hyperion had 220 spectral bands from
400 to 2500 nm, providing continuous, 10 nm spectral resolution. Another long high spectral resolution temporal
record is provided by the European Space Agency's Compact High Resolution Imaging Spectrometer (CHRIS), able
to acquire up to 63 spectral bands from 400–1050 nm. CHRIS was launched aboard Proba-1 on 22 October 2001
and is operational at present. The Naval Research Laboratory had the first water focused hyperspectral sensor, the
Hyperspectral Imager for the Coastal Ocean (HICO) on the International Space Station with more than 80 bands and
providing 5 years of data (September 2009–2014) (Corson and Davis, 2011). The Italian Space Agency launched
the Hyperspectral Precursor of the Application Mission (PRISMA) mission in March 2019. Germany plans to
launch the Environmental Mapping and Analysis Program (EnMAP) upon completion of Phase D, notionally in
2020. Ongoing airborne missions of high spectral resolution capabilities include instruments such as NASA's
Airborne Visible-Infrared Imaging Spectrometer-Next Generation (AVIRIS-NG) and an airborne hyperspectral
sensor, HyMap. Many other high spectral resolution satellite and airborne missions are in recent operation or



deployment stages, such details can be gleaned for example from the Committee on Earth Observation Satellites (CEOS) database (http://database.eohandbook.com).

NASA's Plankton, Aerosol, Cloud, ocean Ecosystem (PACE) satellite mission is intended to be a hyperspectral atmospheric and ocean color mission to be launched in 2022–2023 and to provide data to further the understanding

of a myriad of Earth system processes including those involving ocean ecology, biogeochemistry, as well as atmospheric composition and dynamics (see more details in Werdell et al., 2019). One of the central objectives of the PACE mission is to improve our understanding and quantification of the aquatic biogeochemical cycling and ecosystem function in response to anthropogenic and natural environment variability and change. High spectral resolution coincident IOP/AOP data are required to aid in development and refinement of algorithms to characterize

and quantify aquatic conditions and for the calibration and validation of satellite measurements. A Surface Biology and Geology mission is an additional likely upcoming U.S. space agency hyperspectral mission. It has been recommended as the first Earth Observation mission to come following the currently scheduled remote sensing missions. This Surface Biology and Geology mission is targeted to collect hyperspectral visible–shortwave infrared imagery and multi- or hyperspectral thermal imagery, at 30–60 m spatial resolution and will include measurements

of inland and coastal environments (National Academies of Sciences, Engineering, and Medicine, 2018).

At present, there is a paucity of coincident in situ optical aquatic measurements of high spectral resolution. There are databases providing multispectral resolution IOPs and AOPs, with varying degrees of updates in recent years (e.g. Werdell and Bailey, 2002; Werdell and Bailey, 2005; Valente et al., 2019). We present the first organization of

existing quality-controlled hyperspectral IOP and AOP data from polar, open ocean, estuary, coastal, and inland water. The dataset is intended for remote sensing algorithm development activities associated with upcoming high spectral resolution satellite and airborne missions.

## 2 Materials and Methods

In 2015, in the early development of the PACE Mission, there was an open call to the aquatic remote sensing community to contribute well-documented, quality-controlled data sets consisting of near-synchronous depth profiles of IOPs and AOPs within the water column and near-surface reflectance and optical properties as part of an international effort to build a dataset for algorithm development and testing. All contributors to the database have actively taken part in the quality assessment of the data. Variable assignments, accuracy estimates, and measurement

details were given and confirmed by the data providers. Data that either had IOP or AOP at high spectral resolution were included in the dataset. To arrange data in an organized, uniform structure, data were edited as follows. Data were filtered by considering depths from the surface to no greater than 50 m depth. We rounded data provided at fractional wavelengths to the nearest integer. Missing data is represented in the data files by placeholder values of −999. Metadata is provided at the top of each data file, detailing the contact information for the data provider, the

file source, data publication reference(s), native data collection range and resolution. The spectral range of the



database is 300–900 nm, provided at 1 nm resolution. Variables included in the database are listed in Table 1. Data collection characteristics are presented in Table 2. Figure 3 and Table 3 detail the global distribution of coincident IOP/AOP data. In general terms, AOPs were measured using commercially available radiometer systems that either float at the surface or vertically profile the water column. IOPs were measured using in-water instrumentation and

spectrophotometric analysis of discrete water samples (i.e. water sample removed from the aquatic environment). Brief descriptions of provider and cruise-specific protocols and methodology are given in the following paragraphs.

## 2.1 Methods by Data Contributor and Expedition

In this section, the data providers describe their specific data collection methods used in acquiring and processing

the provided data. Methods not previously published in peer reviewed literature are detailed fully here.

### 2.1.1 Ackleson – RIO-SFE-1 and RIO-SFE-3

Ackleson provides in situ data from the Remote and In Situ Observations - San Francisco Bay and Delta Ecosystem (RIO-SFE) data collection efforts over nine stations in the bay area of San Francisco, California, USA. In-water

spectral absorption and attenuation were measured using a WETLabs AC-S and AC-9. The AC-9 intake was passed through a 0.7 μm cartridge filter to remove particulates, thus, those measurements represent only very small particles and dissolved impurities ($a_{cdom}$ and $c_{cdom}$). The particulate absorption coefficient, $a_p(\lambda)$, was calculated from the difference between AC-S measurements of whole water, $a(\lambda)$ and ac-9 $a_{cdom}(\lambda)$. Backscattering, $b_b(\lambda)$, was measured by a WETLabs ECO-VSF 3.


Above-water $R_{rs}(\lambda)$ was measured between 400 nm and 900 nm using an Analytical Spectral Devices (ASD; Boulder, CO) handheld Spectrometer. The procedure for measuring reflectance is a modified version of Carder and Steward (1985). At each station, 10 sets of measurements were made consisting of 1) reflected radiance from a Spectralon 10% reflectance plaque (Labsphere, Inc., North Sutton, NH), 2) radiance reflected from the sea surface,

and 3) radiance from the section of the sky that would be reflected off the sea surface at the measurement angle. These repetitions were completed as rapidly as possible in order to minimize the impact of changing light or water conditions. Measurements were made between 90° and 135° azimuthal angle relative to the position of the sun and at a 30° angle relative to the vertical to minimize sun glint (Mobley and Stramski, 1997; Mobley 1999).

### 2.1.2 Boss, Chase – Tara Expeditions and SABOR

The Tara Oceans expedition was a two-and-a-half-year long ocean cruise, intended to provide a sampling of the world's diverse ocean environments. The Tara Oceans Polar Circle Expedition (Tara Arctic) took place from May through December 2013 and allowed collection of data in the Arctic Ocean. The Tara Mediterranean expedition (Tara Med) took place from June through September 2014 in the Mediterranean Sea. The Ship-Aircraft Bio-Optical





Research (SABOR) collaborative research campaign allowed scientists to gather data from the Gulf of Maine, North
       Atlantic and Mid-Atlantic coast from July thru August 2014. A full description of these Tara and SABOR
       expeditions and the Boss/Chase provided IOP and AOP datasets and data processing can be found in Boss et al.,
       2013, Chase et al., 2017, and Matsuoka et al., 2017. Briefly, IOPs were measured by an inline system that included
       a WET-Labs AC-S, a CDOM fluorometer and a thermosalinograph. Particulate properties were computed from the

difference between measurements of the total and dissolved fraction (Dall'Olmo et al., 2009; Slade et al., 2010).
       Absorption by the dissolved fraction was computed by interpolating between daily discrete samples collected with a
       2 m long Ultra Path capillary wave guide using the filtered AC-S measurements (Matsuoka et al., 2017). During the
       Tara Oceans/Mediterranean and the SABOR campaigns, reflectance was measured using a Satlantic hyper-spectral
       radiometer buoy (a.k.a. HyperPro in buoy mode), with radiance measured by the upwelling radiometer and

propagated to the surface using a bio-optical model, and then used together with downwelling irradiance to calculate
       remote-sensing reflectance ($R_{rs}(\lambda)$) (see Chase et al., 2017 for details on data processing). During the Tara Arctic
       campaign, a C-OPS profiling radiometer system was used to measure upwelling radiance and downwelling
       irradiance and subsequently calculate $R_{rs}(\lambda)$ at 19 wavelengths between 320 nm and 880 nm.

**2.1.3 Bricaud – BIOSOPE**

       The BIogeochemistry & Optics SOuth Pacific Experiment (BIOSOPE) cruise on R/V l'Atalante, from October
       through December 2004 followed an 8000 km transect from the mesotrophic waters around the Marquesas Islands to
       the hyperoligotrophic waters of the South Pacific Gyre, and then to the eutrophic waters of the upwelling area off
       Chile. BIOSOPE was a collaborative cruise where participating investigators were responsible for making subsets

of optical measurements. With the combined data of the contributing BIOSOPE investigators, nearly all BIOSOPE
       campaign stations contain complete sets of AOP and IOP data. This section summarizes Bricaud's methodologies in
       BIOSOPE campaign data collection. A detailed description of the dataset and methods can be found in Bricaud et al.
       (2010).

Particulate and CDOM absorption measurements were made on board. For particulate absorption measurements,
       seawater samples were collected on Whatman GF/F filters, and absorption spectra, $a_p(\lambda)$, were measured using the
       filter pad technique (with a soaked blank filter as a reference), using a Perkin-Elmer Lambda-19 spectrophotometer
       equipped with an integrating sphere. Non-algal absorption spectra, $a_{nap}(\lambda)$, were measured on the same filters after
       pigment extraction in methanol (Kishino et al. 1985). When necessary, the residual absorption due to incompletely

extracted pigments was corrected by applying an exponential fit (over the wavelength ranges where pigment
       absorption is negligible) to actual spectra.

       All spectra were shifted to zero in the near infrared (750–800 nm average) to minimize possible differences between
       sample and reference filters. Measured optical densities were corrected for the pathlength amplification effect

(according to Allali et al. 1997 for clear waters, and to Bricaud and Stramski, 1990 for eutrophic waters) and then





converted into absorption coefficients (in m$^{-1}$). Finally, phytoplankton absorption spectra, $a_{ph}(\lambda)$, were obtained by subtracting $a_{nap}(\lambda)$ from $a_p(\lambda)$.

CDOM absorption measurements were performed using a WPI Ultrapath capillary waveguide with a 2 m pathlength. Samples were filtered under dim light into glass bottles, using pre-rinsed 0.2 μm Sartorius filters, and then analysed immediately. High-performance liquid chromatography quality water, artificially salted (35 g L$^{-1}$) with precombusted NaCl, was used as reference water. Between each measurement, the sample cell was cleaned according to the WPI, Inc. recommendations. Replicate measurements (including all handling steps) showed that the reproducibility was approximately ± 0.005 m$^{-1}$ at 375 nm.


### 2.1.4 Craig – BBOMB

All measurements from provider Craig are derived from collection of data at the Bedford Basin Ocean Monitoring Buoy (BBOMB), a coastal ocean monitoring buoy located in the Bedford Basin near Halifax, Nova Scotia, Canada. A full description of the Craig dataset and acquisition protocols can be found in Craig et al. (2012). Water samples
were collected by Niskin bottle at a depth of 1 m for the determination of various water column parameters, that included spectral particulate absorption coefficient, $a_p(\lambda)$ and $a_{cdom}(\lambda)$. Wherever possible, NASA Ocean Optics Protocols (Pegau et al., 2003) were followed for all sample acquisition, handling, storage and analysis. Briefly, $a_p(\lambda)$ and $a_{ph}(\lambda)$ spectra were determined from water samples that were filtered under low pressure through a 25 mm GF/F (Whatman) filter. The particulate absorption coefficient, $a_p(\lambda)$, in the range 350–800 nm was determined in a Cary
UV–VIS spectrophotometer with the filter pad mounted on a quartz glass slide and placed at the entrance to an integrating sphere in a modification (Craig, 1999) of the Shibata (1959) opal glass technique. Samples were de-pigmented by soaking the filters in a 0.1% active chlorine solution of NaClO (Kishino et al., 1985; Tassan and Ferrari, 1995). The absorption spectra of the de-pigmented particles, $a_{nap}(\lambda)$, were then measured as described above and $a_{ph}(\lambda)$ calculated from $a_p(\lambda) - a_{nap}(\lambda)$.


Depth profiles of hyperspectral downwelling irradiance, $E_d(\lambda, z)$ (μW cm$^{-2}$ nm$^{-1}$) and upwelling radiance, $L_u(\lambda, z)$ (μW cm$^{-2}$ nm$^{-1}$ sr$^{-1}$) (where $z$ is depth in the water column) were made with a HyperPro (Satlantic Inc.) profiling radiometer. Multiple casts (usually three) were made in quick succession and ~100 m away from the boat to avoid the influence of ship shadow (Mueller et al., 2003). A deck unit mounted to the superstructure of the boat also
provided contemporaneous measurements of above-water surface incident irradiance, $E_s(\lambda)$, during profile acquisition.

### 2.1.5 Lewis – BIOSOPE

Another participating science investigator on the BIOSOPE campaign was M. Lewis. This section details his
collection of BIOSOPE cruise data. Remote sensing spectral reflectance ($R_{rs}(\lambda)$, sr$^{-1}$, specifically, the ratio of water-



leaving radiance to downwelling irradiance above sea surface) in the South Pacific gyre was computed from direct measurements of downwelling irradiance above the sea surface ($E_s(\lambda)$, W m$^{-2}$ nm$^{-1}$) taken aboard ship, and measurements of upwelling radiance ($L_u(\lambda)$, W m$^{-2}$ nm$^{-1}$ sr$^{-1}$) made at a depth of 20 cm below the ocean surface, using a modified hyperspectral profiling radiometer adapted to float at the sea-surface and tethered such that the

instrument operated at a distance of ~100 m from the vessel (HyperPro, Satlantic; Claustre et al., 2008; Stramski et al., 2008; Lee et al., 2010). Instrument tilt was measured directly; measurements were rejected if tilts exceeded 5 degrees. Measurements were made over the spectral region 380–800 nm with a resolution of 3.3 nm and with each band having a half-maximum bandpass of 10 nm. Dark values were taken every five samples by use of an internal shutter. These were linearly interpolated for each light value, and then subtracted from the observations.

Calibration coefficients and corrections for immersion effects were obtained following standard protocols (Mueller et al., 2003) and applied to the measurements; demonstrated absolute accuracies are < 2.8% for radiance and < 2.1% for irradiance (see Gordon et al., 2009). Irradiance and radiance data were taken for 3 minutes at each deployment, with each observation within the deployment time-series representing integration times of 0.03 to 0.5 seconds, depending on the intensity of the incident radiance. These measurements were then interpolated to a common time

frame at a frequency of every 2 seconds and to a common spectral resolution every 2 nm.

Upwelling radiance measurements were then propagated to the sea-surface using an iterative approach that estimates the spectral diffuse attenuation coefficient from spectral ratios of measured radiance, and the water-leaving radiance above the sea surface, $L_W(\lambda)$, is then computed based on Fresnel reflectance at the water–air boundary and the real

relative index of refraction of water (Mueller et al., 2003). A 3 minute time series of $R_{rs}$ was made by dividing the computed water-leaving radiance by the downward irradiance for each time interval, and an average value and standard deviation computed for each deployment.

### 2.1.6 Mouw – Lake Superior Studies

Provider Mouw contributed data from measurements made by researchers at the University of Rhode Island in the inland water body, Lake Superior, the largest of the Great Lakes of North America. A detailed description of the methods used for inland IOP and AOP observations can be found in Mouw et al. (2017). Optical and biogeochemical data were collected in Lake Superior during the ice-free months (May–October) of 2013 through 2016. The dataset consists of a full suite of coincident IOPs and AOPs, including $a$, $a_{cdom}$, $a_{cdom\_dis}$, $a_{nap\_dis}$, $a_{nw}$, $a_p$,

$a_{p\_dis}$, $b_b$, $b_{bp}$, $c$, $c_{nw}$, and $R_{rs}(\lambda)$. The variables used to retrieve $R_{rs}$ are available by request from the data contributor. The contributor also notes that $a_{ph}$ can be calculated from the provided variables.

AOP radiometric measurements were made with three HyperOCR spectral radiometers (Satlantic Inc.) that measure between 350 nm and 800 nm with approximately 3 nm resolution (137 total wavelengths). In-water $E_d(\lambda)$ and $L_u(\lambda)$

HyperOCR sensors were attached to a free-falling Profiler II frame (Satlantic Inc.), while the $E_s(\lambda)$ sensor was mounted on top of the ship to allow for correction of the other measurements due to changing sky conditions. At



each station, the system was deployed for three cast types: surface, multi- and full profile. To characterize the air–water interface, a floatation collar on the profiler frame enabled continuous measurement of $L_u(\lambda)$ approximately 20 cm below the water surface for 5 minutes (surface profile). The flotation collar was removed, and the profiler then

deployed in free-fall mode, measuring five consecutive profiles from the surface to 10 m to characterize the near-surface light field (multi-profile). Finally, the profiler was allowed to free-fall to the 1% light level or to within 10 m of the bottom, whichever was shallower (full profile). All methods and analysis follow the NASA ocean optics protocols for satellite ocean color sensor validation (Mueller et al., 2003).

IOPs were collected via a vertically profiled bio-optical package that measures absorption, attenuation (WET Labs AC-S) and backscattering (WET Labs ECO-BB9) along with concurrent temperature, salinity (SeaBird CTD 37SI) and fluorometeric chlorophyll $a$ (WET Labs ECO-FL3). All methods and analysis followed the NASA ocean optics protocols for satellite ocean color sensor validation (Mueller et al., 2003). Total absorption and attenuation ($a(\lambda)$ and $c(\lambda)$, m$^{-1}$, respectively) were resolved at 81 wavelengths between 400–750 nm.


For laboratory analysis of discrete water samples, spectral CDOM, particulate, non-algal and phytoplankton absorption were measured spectrophotometrically (Perkin-Elmer Lambda 35 UV/VIS dual-beam) for wavelengths between 300 nm and 800 nm. Absorption of CDOM filtrate was measured in a 10 cm cuvette following NASA's Ocean Optics Protocols (Mueller et al., 2003) using a slit-width of 2 nm and a scan rate of 240 nm min$^{-1}$. For

particulate and non-algal absorption, we followed the transmission–reflectance ($T$–$R$) method (Tassan and Ferrari, 2002; Lohrenz, 2000; Lohrenz et al., 2003) that utilizes an integrating sphere to correct measurements for the contribution of scattering.

### 2.1.7 Schaeffer – Florida Estuary Optics

Provider Schaeffer collected in situ measurements and water samples during boat-based surveys in Florida estuaries between September 2009 and November 2011. Hydrographic profiling measurements were collected using a Seabird CTD package. A free-falling hyperspectral profiling system (HyperPRO, Satlantic, Halifax, NS, Canada) provided in-water hyperspectral (400–735 nm, interpolated every 1 nm) measures of downwelling irradiance ($E_d(z,\lambda)$), upwelling radiance ($L_u(z,\lambda)$), and depth ($z$). Water samples were collected 0.5 m below the air-water

surface for absorption (phytoplankton pigment, non-algal particles, CDOM) and extracted chlorophyll analyses. CDOM absorption was measured in a 10 cm cuvette using a Shimadzu UV1700 dual-beam spectrophotometer at 1 nm intervals between 200–700 nm with Milli-Q deionized water as a reference. Total particulates were collected on Whatman 25 mm GF/F filters and analyzed with a Shimadzu UV1700 dual-beam spectrophotometer at 1 nm intervals between 400–800 nm with 0.2 μm filtered seawater as the reference standard (Pegau et al., 2003).

Pigments were extracted from filters with warm methanol and rescanned to measure the detrital absorption (Kishino et al., 1985).





Remote-sensing reflectance ($R_{rs}$) was derived from both a profiling radiometer (HyperPro, Satlantic) and a hyperspectral surface acquisition system (HyperSAS, Satlantic Inc., Halifax, Nova Scotia). The HyperSAS logged spectral measurements of above-water radiance ($L_t(\lambda)$), sky radiance ($L_i(\lambda)$), and downwelling sky irradiance ($E_s(\lambda)$) from 350 to 800 nm (interpolated at 1 nm intervals). The above-water remote-sensing reflectance spectra were corrected, following the surface correction algorithm of Gould et al. (2001), using the average absorption at 412 nm and the derived spectral scattering shape (Gould et al., 1999). Florida Estuaries archived data is available at https://doi.org/10.23719/1424031.

### 2.1.8 Stramski, Reynolds – BIOSOPE, ANT26, KM12

Stramski and Reynolds provide data for three cruises, BIOSOPE, ANT26 and KM12. ANT26 was a German cruise onboard the R/V Polarstern, covering a south-to-north segment of the Atlantic Ocean from Punta Arenas, Chile area (beginning in April 2010) to Bremerhaven, Germany area (finishing in May 2010). The KM12 cruise collected data in the Pacific Ocean off the Hawaiian Islands in June 2012. For all three cruises, the spectral backscattering coefficient of seawater, $b_b(\lambda)$, was measured in situ from vertical profiles obtained with a combination of HOBI Labs Hydroscat-6 and a-$\beta$eta sensors. The determination of $b_b(\lambda)$, and the particulate contribution $b_{bp}(\lambda)$ from these measurements is described in Stramski et al. (2008) and Zheng et al. (2014). On BIOSOPE, a Hydroscat-6 providing measurements at six wavelengths (442, 470, 550, 589, 620, and 671 nm) was paired with two single wavelength a-$\beta$eta sensors (420 and 510 nm). For the ANT26 and KM12 cruises, a combination of two Hydroscat-6 instruments was used to provide measurements in eleven spectral bands (394, 420, 442, 470, 510, 532, 550, 589, 640, 730, and 852 nm; 550 nm common to both instruments).

For the ANT26 and KM12 cruises, discrete water samples within the upper 5 m were collected from a CTD-Rosette equipped with Niskin bottles. The spectral absorption coefficient of particulate material, $a_p(\lambda)$, was determined spectrophotometrically with a filter pad technique for particles retained on a 25 mm glass fiber filter (GF/F, Whatman). Measurements were made at 1 nm resolution over the spectral region 300–850 nm using a Perkin-Elmer Lamba 18 spectrophotometer equipped with a 15 cm diameter integrating sphere. The filters were placed inside the sphere to minimize potential scattering error, and the correction for pathlength amplification factor determined for this configuration of measurement was used (Stramski et al., 2015). The partitioning of $a_p(\lambda)$ into phytoplankton, $a_{ph}(\lambda)$, and non-algal particle, $a_{nap}(\lambda)$, contributions was accomplished through the chemical extraction of pigments using methanol (Kishino et al., 1985). The absorption coefficient of CDOM, $a_{cdom}(\lambda)$, on ANT26 was determined on discrete water samples using a PSICAM instrument (Röttgers and Doerffer, 2007). For KM12, $a_{cdom}$ was measured in situ using a WET Labs AC-S.

The spectral remote-sensing reflectance, $R_{rs}(\lambda)$, for the ANT26 and KM12 cruises was determined by averaging a time-series of radiometric measurements from a Satlantic HyperPro II radiometer attached to a surface float and deployed at a large distance from the vessel. Measurements were obtained over the spectral range 350–800 nm at



~3 nm resolution and subsequently interpolated to 1 nm intervals. Subsurface measurements of the upwelling zenith
radiance (i.e., light propagating towards zenith) made at 0.2 m depth were propagated to and across the sea-surface
and combined with above-surface measurements of downwelling planar irradiance to estimate $R_{rs}(\lambda)$ (Uitz et al.,
2015).

**3 Data Availability**

The diverse set of in situ apparent and inherent optical property data are stored and provided free of charge at the
PANGAEA data archive and publisher for Earth and Environmental Science. Data are available as Microsoft Excel
(.xlsx) files. The primary link for accessing the data is https://doi.pangaea.de/10.1594/PANGAEA.902230 (Casey et
al., 2019). Individual variable files are stored and available via interactive HTML download at
https://doi.pangaea.de/10.1594/PANGAEA.902230?format=html#download or via tab-delimited text at
https://doi.pangaea.de/10.1594/PANGAEA.902230?format=textfile.

**4 Results and Discussion**

Overall, the collection of datasets provides mostly coincident IOP/AOP data from a wide range of latitudes and
water types, including polar, open ocean, estuary, coastal and inland water environments. We detailed the specific
cruise, instrument and methodology approaches taken by each data provider. The majority of the data has been
published as referenced. The few contributed data sets which are not yet published in peer reviewed literature are
fully described in this manuscript. Thus, the data provide a robust means to evaluate aquatic remote sensing
observations toward further remote sensing science research and development goals. The in situ dataset has been
stored and is provided free of charge at the PANGAEA data archive and publisher for Earth and Environmental
Science (https://doi.pangaea.de/10.1594/PANGAEA.902230) as detailed in Section 3.

Hereafter we describe the spatial and temporal resolution covered by the dataset for coincident IOP and AOP, where
when referring to coincident data, we describe data that have $R_{rs}$ and at least one IOP variable available. IOP and
AOP data are provided from 12 cruises, from 2004 through 2016, covering Arctic, mid- and equatorial open ocean,
estuary, coastal and inland aquatic sites (see Table 2, Fig. 3). A summary of the number of data points available for
every cruise for each of the variables is provided in Table 3. Table 3 shows that IOPs are generally collected at more
stations than AOPs. The number of data available for IOPs is also much larger than AOPs because we count every
single depth as a single data point. The three datasets with the largest amount of data (where each station, depth and
variable count as a data point) were those provided by Ackleson, Mouw and Stramski/Reynolds. The Ackleson
dataset contains data for seven IOPs and $R_{rs}$, Mouw provides data for nine IOPs as well as $R_{rs}$ and the
Stramski/Reynolds dataset has data for six IOPs plus $R_{rs}$. Data from the largest datasets are also geographically
diverse. Specifically, the Stramski/Reynolds dataset includes between 21 and 57 different geographic stations
depending on the IOP data variable, the Mouw dataset includes between 63 and 102 different geographic stations





and the Ackleson dataset provides between nine and 33 different geographic stations. Of note, from the BIOSOPE

collaborative cruise, coincident BIOSOPE AOP and IOP data is provided by a suite of contributors. In this manuscript, BIOSOPE data contributors and variables include Bricaud ($a_{cdom}$, $a_{nap}$, $a_p$), Lewis ($R_{rs}$) and Stramski/Reynolds ($b_b$).

Similar to the synergies of the BIOSOPE campaign with multiple investigators, dataset users are also encouraged to

consider harnessing provided data to derive additional desired variables. For example, many stations contain a complete set of both an AOP measurement ($R$ or $R_{rs}$) and the two main IOPs ($a$ and $b_b$). Note that total absorption can be calculated if all constituent absorption coefficients are measured in conjunction with published IOPs of pure water; this applies to most of our stations. Derivation and combinations of provided data ultimately depend on the intent and goals of the user.


We show the range in reflectance values ($R_{rs}$) provided from diverse geographic locations of inland, estuary, coastal, open ocean and polar waters in Fig. 4. The diversity in the signal from inland water (Fig. 4, f) to coastal (Fig. 4, a, d, g), Arctic (Fig. 4, b) and open ocean waters (Fig. 4, c, e, h) show a range of the various particulate, biogeochemical, and other water conditions characteristic of different aquatic environments. The graphs also show

the level of detail that can be extracted by varying spectral resolutions. Lower spectral resolution is shown in the Boss/Chase Arctic data (from 5 nm to tens of nm separation), and higher resolution is found in Boss/Chase Tara Oceans, Lewis and Mouw (2–3 nm spectral resolution) and Ackleson, Craig, Schaeffer, and Stramski/Reynolds (1 nm spectral resolution).

When assessing the geographic distribution of the coincident IOP and AOP data, we found data were more frequent for latitudes between 30°N and 40°N and longitudes between 50°W and 100°W (Fig. 5). The Ackleson (San Francisco Bay), Schaeffer (northern Gulf of Mexico) and Mouw (Lake Superior) data were acquired at those latitudes and longitudes. These results also highlight the lack of data for the area between 100°E and 180°E and at latitudes south of 50°.


We caution users of the datasets to consider inherent limitations to certain data collections. For example, some data collected in turbid waters were found to contain less signal compared to noise. Specifically, the AC-S dataset of Boss/Chase has significant uncertainties in $a_p$ in the blue part of the spectrum due to uncertainty in the scattering correction of this measurement, particularly in turbid waters (e.g. Stockley et al., 2017). Additionally, as previously

detailed, not all data collected is coincident. We have indicated in Fig. 3 and Table 3 and several details including the geographic and variable distribution concerning coincident data. Overall, because most data have already been published in peer review literature with study collection, processing and analysis details and where needed fully detailed here; readers are able to determine the utility and applicability of the datasets provided toward further use of the data.




## 5 Summary

We have compiled aquatic data from a variety of inland, coastal, estuary, and open ocean equatorial, mid- and high-latitude locations. This compilation of aquatic data is a first step in achieving a global distribution of high spectral resolution IOP and AOP data, which we encourage the community to use for aquatic remote sensing algorithm
development and related activities. We recommend further in situ campaigns be commissioned to collect coincident high spectral resolution IOP and AOP data over regions with limited current coverage, for example, high latitude, inland and southerly waters. Such data could also be collected via and in conjunction with upcoming airborne high spectral resolution remote sensing campaigns. Additional in situ data collection over gap areas would be helpful in development, calibration and validation of global algorithms.


As additional high spectral resolution IOP/AOP data become available, this dataset can be expanded accordingly. A comprehensive collection of hyperspectral IOP/AOP datasets would be extremely useful for both development of aquatic remote sensing algorithms, and for the planning of future field sampling missions to address identified gaps. Future expansion of this collection of datasets, beyond addition of optical data, could be inclusion of
biogeochemical information (e.g. phytoplankton pigments, carbon stocks, turbidity, particulate size distribution, and phytoplankton composition) to further assist in development of algorithms relating to biogeochemical parameters. It is crucial to collect coincident high spectral resolution IOP and AOP remote sensing data for the development of robust algorithms. These data, algorithms and scientific investigations can improve our understanding of Earth system biogeochemical, ecological and physical processes on local to global scales.


### Author Contribution

The initial concept for this effort came from the NASA PACE Mission early science project discussions on a goal to provide the community with high spectral resolution datasets. CR and EB sent initial community requests for in situ aquatic data contributions. KC gathered and stewarded the data organization effort. KC led preparation, writing of
the manuscript and generation of figures and tables. All data providers assisted in writing the methods of their data collection. All co-authors contributed to the scientific discussion, review and editing of the manuscript.

### Competing interests

The authors declare that they have no conflicts of interest.

**Acknowledgements**

Collection of in situ aquatic data is a significant time and cost intensive effort. We gratefully acknowledge all data contributors, personnel and agencies involved in support of data collection, processing and provision. EB and AC specifically acknowledge assistance by the Tara Oceans Consortium, Coordinators, Tara Oceans Expedition, Tara





crew, and participants during the Tara Oceans and Tara Polar Circle expedition. We also specifically acknowledge
the BIOSOPE project, led by H. Claustre and funded by the Centre National de la Recherche Scientifique (CNRS),
the Institut des Sciences de l'Univers (INSU), the Centre National d'Etudes Spatiales (CNES), the European Space
Agency (ESA), the National Aeronautics and Space Administration (NASA) and the Natural Sciences and
Engineering Research Council of Canada (NSERC) for the successful cooperative campaign efforts. We thank
Shankar N. Ramaseri Chandra of KBR, contractor to the U.S. Geological Survey Earth Resources Observation and
Science Center for assistance with Figs. 1 and 2.

**Financial Support**

KC and CR were supported by the NASA PACE Project NNX15AE81G. EB and AC data collection, processing and
analysis was supported by NASA grants NNX11AQ14G, NNX09AU43G, NNX15AC08G, NNX13AC42G and
NNX13AE58G. SC data collection was supported by the U.S. Office of Naval Research, the NSERC Research
Partnerships with Satlantic Inc. and Discovery Grants programme, the Government Related Initiatives Program of
the Canadian Space Agency, and Fisheries and Oceans Canada. CM data collection was supported by NASA grant
NNX14AB80G. The collection, processing, and analysis of data submitted by DS and RR was supported by NASA
grants NNG04GO02G, NNX09AK17G, NNX15AC55G, NNX15AQ53G, and ONR grant N000014-09-1-1053. BS,
involved in the Florida Estuaries project, was funded by NASA grant NNH08ZDA001-DECISIONS to the U.S.
EPA Gulf Ecology Division, the USEPA Office of Research and Development, and by EPA Pathfinder Innovation
Grant. This article has been subjected to review by the ORD National Exposure Research Laboratory and approved
for publication. Mention of trade names or commercial products does not constitute endorsement or
recommendation for use. The views expressed in this article are those of the authors and do not necessarily reflect
the views or policies of the US Government.

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



| Variable | Definition, Units |
|---|---|
| $a(\lambda)$ | Total absorption equal to sum of particulate absorption, CDOM absorption and water absorption (m$^{-1}$) |
| $a_{cdom}(\lambda)$ | Colored dissolved organic matter (CDOM) absorption coefficient (m$^{-1}$) |
| $a_{cdom\_dis}(\lambda)$ | Discrete CDOM absorption coefficient (m$^{-1}$) |
| $a_{cdom\_int}(\lambda)$ | Interpolated CDOM absorption coefficient (m$^{-1}$) |
| $a_{cdom\_unc}(\lambda)$ | Uncertainty in measured CDOM absorption coefficient (m$^{-1}$) |
| $a_{nap}(\lambda)$ | Non-algal particle absorption coefficient (m$^{-1}$) |
| $a_{nap\_dis}(\lambda)$ | Discrete non-algal particle absorption coefficient (m$^{-1}$) |
| $a_{nw}(\lambda)$ | Measured absorption with pure water subtracted (m$^{-1}$) |
| $a_p(\lambda)$ | Particulate absorption coefficient (m$^{-1}$) |
| $a_{p\_dis}(\lambda)$ | Discrete particulate absorption coefficient (m$^{-1}$) |
| $a_{p\_unc}(\lambda)$ | Uncertainty in particulate absorption coefficient (m$^{-1}$) |
| $a_{ph}(\lambda)$ | Phytoplankton absorption coefficient (m$^{-1}$) |
| $b_b(\lambda)$ | Total backscattering coefficient (m$^{-1}$) |
| $b_{bp}(\lambda)$ | Particulate backscattering coefficient (m$^{-1}$) |
| $c(\lambda)$ | Total attenuation equal to sum of particulate attenuation, CDOM attenuation and water attenuation (m$^{-1}$) |
| $c_{nw}(\lambda)$ | Measured attenuation with pure water subtracted (non-water attenuation) (m$^{-1}$) |
| $c_p(\lambda)$ | Particulate attenuation coefficient (m$^{-1}$) |
| $c_{p\_unc}(\lambda)$ | Uncertainty in particulate attenuation coefficient (m$^{-1}$) |
| $R(\lambda)$ | Irradiance reflectance (dimensionless) |
| $R_{stdv}(\lambda)$ | Standard deviation of irradiance reflectance (dimensionless) |
| $R_{rs}(\lambda)$ | Remote-sensing reflectance (sr$^{-1}$) |
| $R_{rs\_stdv}(\lambda)$ | Standard deviation of remote-sensing reflectance (sr$^{-1}$) |

**Table 1. List of variables included in the database. Variables are provided as defined by contributor. The symbol '$\lambda$' represents light wavelength in vacuum, in units of nanometer (nm). The term 'discrete' is used to indicate a water sample removed from the aquatic environment.**





| Contributor | Cruise | Location | Collection dates | Parameters measured / derived | Collection instrument(s) | Publications |
|---|---|---|---|---|---|---|
| Ackleson | RIO-SFE-1 RIO-SFE-3 | San Francisco Bay | 5/2014; 3/2015 | $a(\lambda)$, $a_{cdom}(\lambda)$, $a_{cdom\_int}(\lambda)$, $a_{nw}(\lambda)$, $b_b(\lambda)$, $c(\lambda)$, $c_{nw}(\lambda)$, $R_{rs}(\lambda)$, $R_{rs\_stdv}(\lambda)$ | WET Labs spectral ac meters (AC-S, AC-9); ASD Fieldspec HandHeld | Freeman et al., 2017 |
| Boss, Chase | Tara Oceans / Tara Med / SABOR | Atlantic, Indian, Pacific, Mediterranean Sea, coastal and open ocean waters | 1/2010– 3/2012; 6/2014– 9/2014 | $a_p(\lambda)$, $a_{p\_unc}(\lambda)$, $c_p(\lambda)$, $c_{p\_unc}(\lambda)$, $R_{rs}(\lambda)$, $R_{rs\_stdv}(\lambda)$ | AC-S, HyperPro in buoy mode | Boss et al., 2013, Chase et al., 2017 |
| Boss, Chase | Tara Arctic | Arctic coastal and open ocean waters | 5/2013– 10/2013 | $a_{cdom}(\lambda)$, $a_{cdom\_unc}(\lambda)$, $a_p(\lambda)$, $a_{p\_unc}(\lambda)$, $c_p(\lambda)$, $c_{p\_unc}(\lambda)$, $R_{rs}(\lambda)$, $R_{rs\_stdv}(\lambda)$ | AC-S, C-OPS profiling radiometer | Chase et al., 2017; Matsuoka et al., 2017 |
| Bricaud | BIOSOPE | South East Pacific Ocean | 10/2004– 12/2004 | $a_{cdom}(\lambda)$, $a_{nap}(\lambda)$, $a_p(\lambda)$ | WPI Ultrapath; Perkin-Elmer Lambda 19 spectrophotometer with integrating sphere | Claustre, et al., 2008; Bricaud et al., 2010 |
| Craig | BBOMB | Coastal Northwest Atlantic | 2/2009– 3/2010 | $a_{nap}(\lambda)$, $a_p(\lambda)$, $a_{ph}(\lambda)$, $R_{rs}(\lambda)$ | HyperPro, Cary 4000 UV-vis spectrophotometer | Craig et al., 2012 |
| Lewis | BIOSOPE | South East Pacific Ocean | 1/2004– 12/2004 | $R(\lambda)$, $R_{stdv}(\lambda)$, $R_{rs}(\lambda)$, $R_{rs\_stdv}(\lambda)$ | HyperPro in buoy mode | Claustre, et al., 2008; Stramski et al., 2008; Lee et al., 2010 |
| Mouw | Several studies in Lake Superior | Lake Superior, USA | 6/2013– 5/2016 | $a(\lambda)$, $a_{cdom}(\lambda)$, $a_{cdom\_dis}(\lambda)$, $a_{nap\_dis}(\lambda)$, $a_{nw}(\lambda)$, $a_p(\lambda)$, $a_{p\_dis}(\lambda)$, $b_b(\lambda)$, $b_{bp}(\lambda)$, $c(\lambda)$, $c_{nw}(\lambda)$, $R_{rs}(\lambda)$ | HyperOCR spectral radiometers, WET Labs AC-S, WET Labs ECO-BB9, WET Labs ECO-FL3, SeaBird CTD 37SI, Perkin Elmer Lambda 35 | Mouw et al., 2017 |
| Schaeffer | Florida Estuary | Northern Gulf of Mexico | 9/2009– 11/2011 | $a_{cdom}(\lambda)$, $a_{nap}(\lambda)$, $a_p(\lambda)$, $a_{ph}(\lambda)$, $R_{rs}(\lambda)$ | HyperSAS, HyperPRO in | Astuti et al., 2018; Conmy et al., |



| | | | | | |
|---|---|---|---|---|---|
| Optics | estuaries, USA | | | profiling mode, Shimadzu UV1700 | 2017; Keith et al., 2014; Keith et al., 2016; Le et al., 2015; Le et al., 2016; Mishra et al., 2014; Schaeffer et al., 2015 |
| Stramski, Reynolds | BIOSOPE, ANT26, KM12 | South East Pacific Ocean, Atlantic Ocean, Tropical Pacific Ocean | 10/2004–12/2004; 4/2010–5/2010; 6/2012 | $a_{cdom}(\lambda)$, $a_{nap}(\lambda)$, $a_p(\lambda)$, $a_{ph}(\lambda)$, $b_b(\lambda)$, $b_{bp}(\lambda)$, $R_{rs}(\lambda)$ | Perkin-Elmer Lambda 18 spectrophotometer with integrating sphere; HOBI Labs HS-6 or a-βeta; Satlantic HyperPro II in buoy mode — Stramski et al., 2008; Uitz et al., 2015; Loisel et al., 2018 |

**Table 2. In situ data collection instrument, logistical details and related references.**


| | $a$ | $a_{cdom}$ | $a_{cdom\_dis}$ | $a_{cdom\_int}$ | $a_{cdom\_unc}$ | $a_{nap}$ | $a_{nap\_dis}$ | $a_{nw}$ | $a_p$ | $a_{p\_dis}$ | $a_{p\_unc}$ | $a_{ph}$ | $b_b$ | $b_{bp}$ | $c$ | $c_{nw}$ | $c_p$ | $c_{p\_unc}$ | $R$ | $R_{stdv}$ | $R_{rs}$ | $R_{rs\_stdv}$ |
|---|---|---|---|---|---|---|---|---|---|---|---|---|---|---|---|---|---|---|---|---|---|---|
| **RIO-SFE-1,-3** | 22671 (9) | 8831 (34) | | 22671 (9) | | 22671 (9) | | | | | | | 22671 (9) | | 22671 (9) | 22671 (9) | | | | | 9 (9) | 9 (9) |
| **Tara Arctic** | | 29 (29) | | 29 (29) | | | | | 29 (9) | 29 (9) | | | | | | | 29 (9) | 29 (9) | | | 29 (9) | 29 (9) |
| **Tara Oceans/Med, SABOR** | | | | | | | | | 103 (101) | 103 (101) | | | | | | | 103 (101) | 103 (101) | | | 103 (101) | 103 (101) |
| **BIOSOPE (Bricaud)** | | 109 (30) | | | | 138 (36) | | | 141 (38) | | | | | | | | | | | | | |
| **BBOMB** | | | | | | | 43 (1) | | 43 (1) | | | 43 (1) | | | | | | | | | 43 (1) | |
| **BIOSOPE (Lewis)** | | | | | | | | | | | | | | | | | | | 67 (66) | 67 (66) | 67 (66) | 67 (66) |
| **Lake Superior** | 2918 (84) | 2597 (63) | 183 (95) | | | 209 (102) | | 2918 (84) | 2462 (62) | 204 (101) | | | 2959 (84) | 2959 (84) | 2922 (84) | 2918 (84) | | | | | 80 (78) | |
| **Florida Estuary Optics** | | 719 (125) | | | | 719 (125) | | | 718 (125) | | | 719 (125) | | | | | | | | | 621 (141) | |
| **BIOSOPE (Stramski, Reynolds), ANT26, KM12** | | 16 (16) | | | | 21 (21) | | | 21 (21) | | | 21 (21) | 57 (57) | 57 (57) | | | | | | | 21 (21) | |

**Table 3: Data provided by each investigator and cruise. The number in the cells indicates the number of data points available. The number between parentheses is the number of stations in the dataset.**




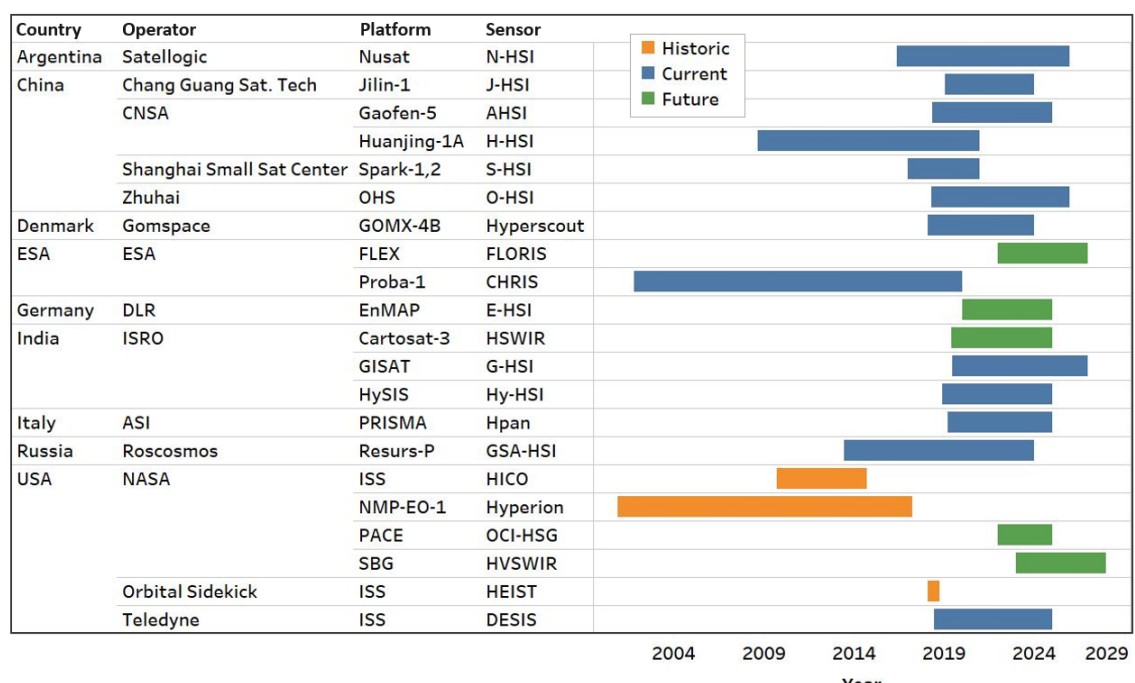

**Figure 1: Illustration of the historical, existing and planned high spectral resolution satellite mission country or agency, operator or agency, platform, sensor name and operation timeline.**





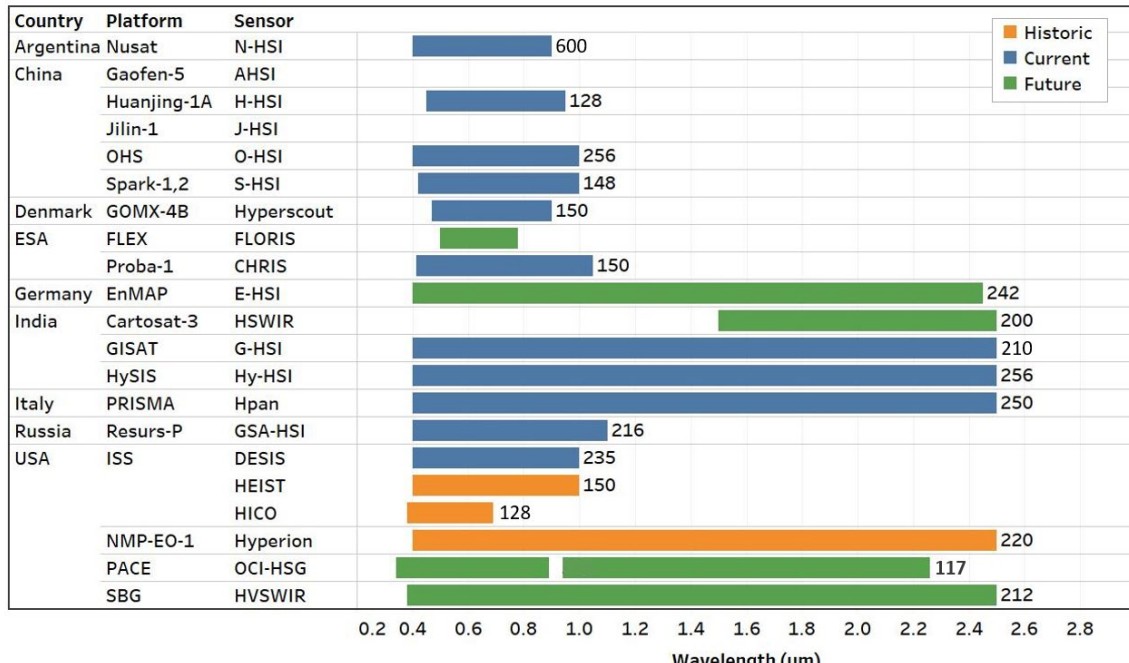

**Figure 2:** Listing of the spectral wavelength range and total number or sensor bands (right of each bar) where presently known for each of the high spectral resolution satellite missions.


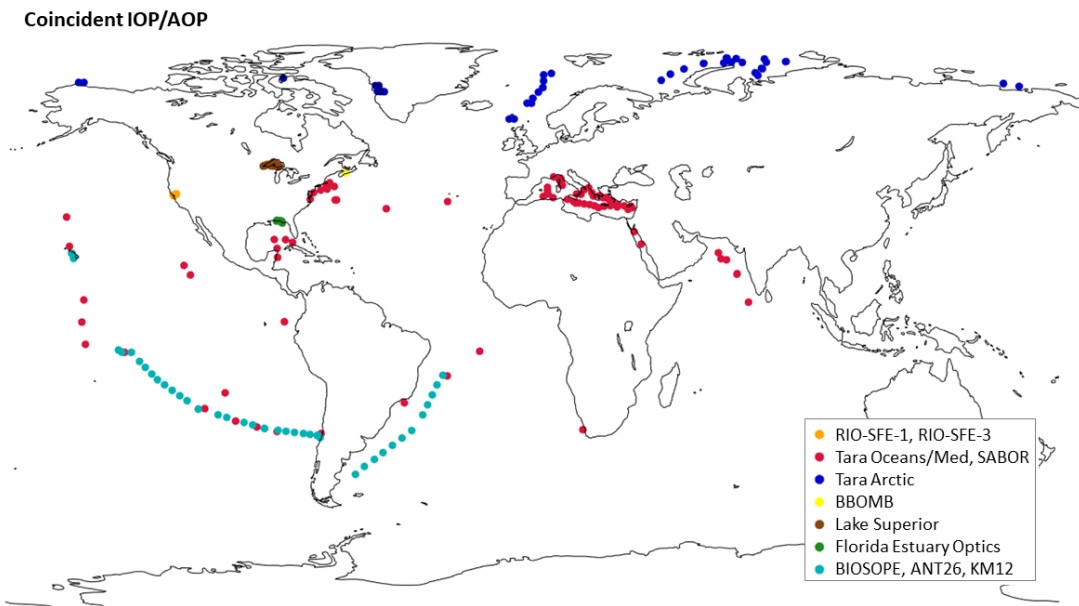

**Figure 3: Global geographic distribution of coincident IOP/AOP data.**


**Figure 4:** Plots demonstrate spectral reflectance diversity of inland, estuary, and ocean environments. Reflectance distribution plots display $R_{rs}$ data from (a.) Ackleson, (b.) Boss/Chase Arctic, (c.) Boss/Chase, (d.) Craig, (e.) Lewis, (f.) Mouw, (g.) Schaeffer and (h.) Stramski/Reynolds. Note, subplots detail the spectral sampling provided; where high spectral resolution data is available, subplots appear more 'linear' (a, d, g, h) and where there is lower spectral resolution data, subplots appear more 'point' (b, c, e, f). Each color in the subplot represents a separate data collection.

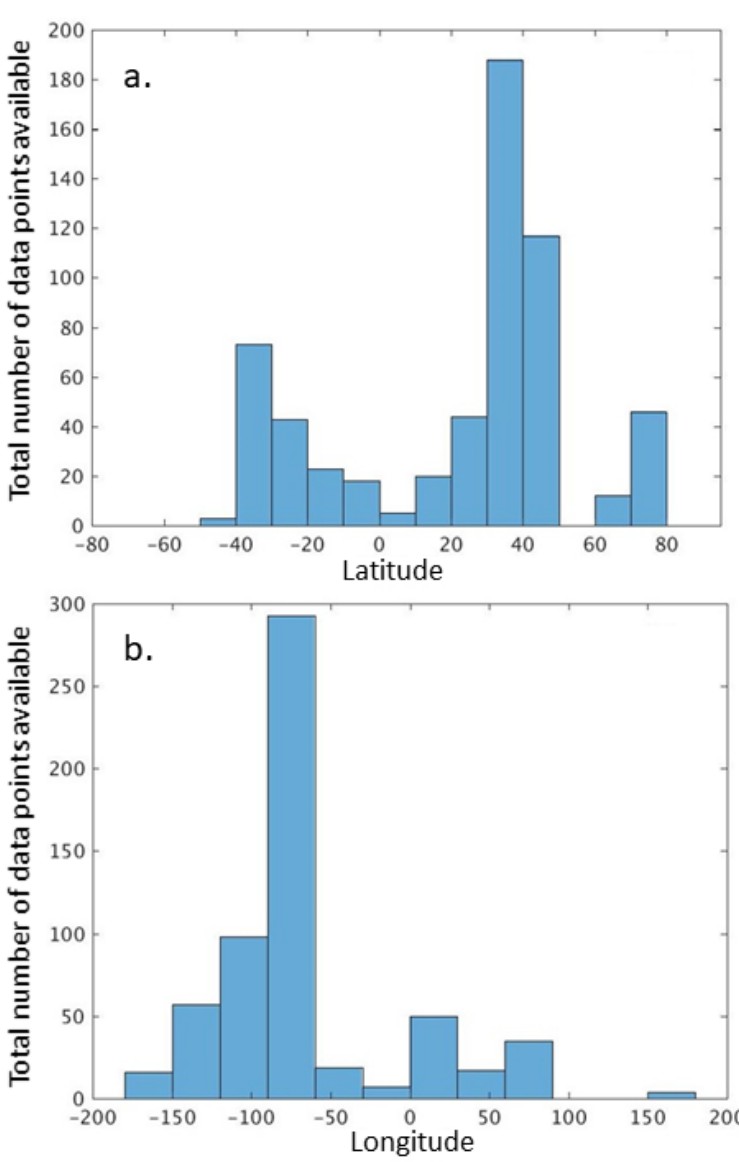


**Figure 5: Geographic frequency distribution of stations with coincident IOP and AOP data. Plot (a) shows data point distribution by latitude and plot (b) shows data point distribution by longitude.**