# Peer review of "A global compilation of in situ aquatic high spectral resolution inherent and apparent optical property data for remote sensing applications"

_Earth System Science Data, 2019_

## Referee Comment (RC1) · Anonymous Referee #1 · 14 Aug 2019

Casey et al compiled a comprehensive AOP/IOP dataset covering wide range of oceanic environments, which I think will be very useful for the ocean optics and ocean color remote sensing community. I fully support its publication.

A few minor suggestions: 1. Line 24, delete "remote sensing" , as it is the same for contact radiometers 2. L69, "water itself" please note that bbw is different between fresh water and seawater, so need to clarify this statement here. 3. L74, "Torrecilla and others", I think it should be Torrecilla et al. 4. Lines 94-111, this summary of current and historical missions is not relevant to this manuscript, could be simply a few citations or references. Also note that some of the missions are/were not designed

for aquatic environments. 5. Line 126, note that Lin et al (2018) have presented a hyperspectral AOP/IOP dataset for the PACE mission. 6. L146, " provided at 1 nm resolution". This could be misleading, as the spectral resolution of many radiometers, including HyperPro, is ∼10 nm. Suggest to change "resolution" to "interval", as it is a simple interpolation of data from much coarser resolution, so not really measurement at 1 nm resolution. This is especially true for bbp, which were usually measured at 6 or 9 bands. 7. L257, "the spectral region 380–800 nm with a resolution of 3.3 nm" From the document of Satlantic, the spectral resolution is about 10 nm, also the sampling interval is 3.3 nm. 8. L265, "a common spectral resolution every 2 nm". Again, it is necessary to be very careful about "resolution", and I think here it is simply a spectral interval of 2 nm for display, not really measured at 2 nm spectral resolution. 9. L326-328, "The above-water remote-sensing reflectance spectra were corrected, following the surface correction algorithm of Gould et al. (2001), using the average absorption at 412 nm and the derived spectral scattering shape (Gould et al., 1999)." Suggest to double check and re-word this approach, as it is not clear how average absorption at 412 nm and derived spectral scattering shape can be used to correct surface reflectance in Rrs measurement. 10. "The in situ dataset has been stored and is provided free of charge at the PANGAEA data archive and publisher for Earth and Environmental Science (https://doi.pangaea.de/10.1594/PANGAEA.902230) as detailed in Section 3" This has been presented earlier, which can be deleted here.

Lin, J., Lee, Z., Ondrusek, M., & Liu, X. (2018). Hyperspectral absorption and backscattering coefficients of bulk water retrieved from a combination of remote-sensing reflectance and attenuation coefficient. Optics Express, 26(2), 157-177

---

## Referee Comment (RC2) · Anonymous Referee #2 · 6 Nov 2019

Casey et al. provide a well-documented and clearly organized dataset of IOP/AOP covering a diverse range of oceanic environments. I encourage the publication of this dataset. It will be extremely beneficial as planned satellite missions become a reality. Just a couple comments to the authors- 1. I suggest paying special attention to your use of "resolution" in the manuscript as it can be misinterpreted in many of its meanings. 2. For Figure 4, consider rearranging subplots to transition geographically, or perhaps organize by the range on the y-axis as it would aid the user/reader in comparing the various regions. Additionally, including a subplot of the map similar to Figure 3 on this Figure would also help with clarity.

---

## Author Response (AR1)

**Compilation of author responses**

Posted author response to review 1

(published 4 October 2019)

*We thank you for your careful review and constructive feedback on the manuscript.  Below we present responses (in regular font italics) to your comments (**in bold**).*

**Minor suggestions:**

**1. Line 24: Delete "remote sensing".**

*Thank you. We agree, we have done this.*

**2. Line 69: "water itself" please note that bbw is different between fresh water and seawater, so need to clarify this statement here.**

*Thank you for this point.  We have edited the sentence to clarify this.*

**3. Line 74: "Torrecilla and others", I think it should be Torrecilla et al.**

*We thank the reviewer for this comment.  This has been changed to Torrecilla et al., 2011 to follow the style guidelines.*

**4. Lines 94-111: this summary of current and historical missions is not relevant to this manuscript, could be simply a few citations or reference.  Also note that some of the missions are/were not designed for aquatic environments.**

*We thank the reviewer for the comment. We agree that some of the missions are not directly geared toward aquatic remote sensing, though note that research scientists and operational data managers are opportunistic in use of available sensor data. For example, a robust set of aquatic remote sensing algorithms and products have been developed using Landsat data, a sensor primarily aimed at land remote sensing.  For this reason, we provided details on key potentially useful high spectral resolution sensors for interested readers.  We have edited the paragraph to reduce the amount of text and narrow the overview of sensors to those especially suited towards high spectral resolution aquatic remote sensing.*

**5. Line 126:  note that Lin et al (2018) have presented a hyperspectral AOP/IOP dataset for the PACE mission.**

**Lin, J., Lee, Z., Ondrusek, M., & Liu, X. (2018). Hyperspectral absorption and backscattering coefficients of bulk water retrieved from a combination of remote-sensing reflectance and attenuation coefficient. Optics Express, 26(2), 157-177**

*We thank you for this information.  The references listed were an example and not meant to be completely exhaustive.  We have added the provided reference in the introduction as suggested.*

**6. Line 146: "provided at 1 nm resolution". This could be misleading, as the spectral resolution of many radiometers, including HyperPro is ~10 nm.  Suggest to change resolution to interval, as it is a simple interpolation of data from much coarser resolution, so not really measurement at 1 nm resolution. This is especially true for bbp, which were usually measured at 6 or 9  bands.**

*We thank you for this point. We agree that for some data sets, the providers or we interpolated data to integer nm format.  We have thus changed "resolution" to "interval" as advised.*

**7. Line 257: "the spectral region 380–800 nm with a resolution of 3.3 nm" From the document of Satlantic, the spectral resolution is about 10 nm, also the sampling interval is 3.3 nm.**

*We thank you for this comment. For clarity, we reworded the sentence as advised listing first the resolution, then the sampling interval.*

**8. Line 265: "a common spectral resolution every 2 nm". Again, it is necessary to be very careful about "resolution", and I think here it is simply a spectral interval of 2 nm for display, not really measured at 2 nm spectral resolution.**

*We thank you for this comment. We have changed the sentence as recommended.*

**9. Lines 326-328: "The above-water remote-sensing reflectance spectra were corrected, following the surface correction algorithm of Gould et al. (2001), using the average absorption at 412 nm and the derived spectral scattering shape (Gould et al., 1999)." Suggest to double check and re-word this approach, as it is not clear how average absorption at 412 nm and derived spectral scattering shape can be used to correct surface reflectance in Rrs measurement.**

*We have revised and added some additional text to briefly describe the steps followed in the Gould et al. 2001 method.*

**10. "The in situ dataset has been stored and is provided free of charge at the PANGAEA data archive and publisher for Earth and Environmental Science (https://doi.pangaea.de/10.1594/PANGAEA.902230) as detailed in Section 3" This has been presented earlier, which can be deleted here.**

*We thank the reviewer for this point. We have removed the redundant sentence (which originally appeared at the end of the 1$^{st}$ paragraph of section 4).*

**Posted author response to review 2**

**(published 25 November 2019)**

*We thank Anonymous Referee #2 for the review and constructive feedback on the manuscript. Below we present responses (in regular font italics) to Anonymous Referee #2 comments (**in bold**).*

**Casey et al. provide a well-documented and clearly organized dataset of IOP/AOP covering a diverse range of oceanic environments. I encourage the publication of this dataset. It will be extremely beneficial as planned satellite missions become a reality.**

**Comments to the authors:**

**1. I suggest paying special attention to your use of "resolution" in the manuscript as it can be misinterpreted in many of its meanings.**

*Thank you for this point. We have thoroughly reviewed the manuscript and paid careful attention to ensure consistent and correct use of the word 'resolution'. We have changed 'resolution' to 'sampling interval' where appropriate, as advised by the other reviewer. We have otherwise clarified sentences where appropriate.*

**2. For Figure 4, consider rearranging subplots to transition geographically, or perhaps organize by the range on the y-axis as it would aid the user/reader in comparing the various regions. Additionally, including a subplot of the map similar to Figure 3 on this Figure would also help with clarity.**

*We thank the reviewer for this comment. To improve user readability, we have added labels with the cruise name as well as location to the graphs and supplemented the Figure 4 caption. We have kept the order of the graphs the same to remain consistent with the article text, tables and other figures in the paper. In the caption, we refer readers to the map provided in Figure 3 (matching location with cruise names). We find that adding another map to Figure 4 may be repetitious and detract from space to display the spectral reflectance information.*

**List of all relevant changes made in the manuscript**

a)  We made all changes detailed in the responses to the reviewer comments above.

b)  We added new hyperspectral mission updates regarding the NASA PACE mission and DESIS mission in Section 1.

**Marked-up manuscript** (follows)

[revised manuscript text omitted]
_{\mathrm{cdom}}(\lambda)$, $a_{\mathrm{cdom\_unc}}(\lambda)$, $a_{\mathrm{p}}(\lambda)$, $a_{\mathrm{p\_unc}}(\lambda)$, $c_{\mathrm{p}}(\lambda)$, $c_{\mathrm{p\_unc}}(\lambda)$, $R_{\mathrm{rs}}(\lambda)$, $R_{\mathrm{rs\_stdv}}(\lambda)$ | AC-S, C-OPS profiling radiometer | Chase et al., 2017; Matsuoka et al., 2017 |
| Bricaud | BIOSOPE | Southeast Pacific ocean | 10/2004–12/2004 | $a_{\mathrm{cdom}}(\lambda)$, $a_{\mathrm{nap}}(\lambda)$, $a_{\mathrm{p}}(\lambda)$ | WPI Ultrapath; Perkin-Elmer Lambda 19 spectrophotometer with integrating sphere | Claustre, et al., 2008; Bricaud et al., 2010 |
| Craig | BBOMB | Coastal Northwest Atlantic ocean | 2/2009–3/2010 | $a_{\mathrm{nap}}(\lambda)$, $a_{\mathrm{p}}(\lambda)$, $a_{\mathrm{ph}}(\lambda)$, $R_{\mathrm{rs}}(\lambda)$ | HyperPro, Cary 4000 UV-vis spectrophotometer | Craig et al., 2012 |
| Lewis | BIOSOPE | Southeast Pacific ocean | 1/2004–12/2004 | $R(\lambda)$, $R_{\mathrm{stdv}}(\lambda)$, $R_{\mathrm{rs}}(\lambda)$, $R_{\mathrm{rs\_stdv}}(\lambda)$ | HyperPro in buoy mode | Claustre, et al., 2008; Stramski et al., 2008; Lee et al., 2010 |
| Mouw | Several studies in Lake Superior | Lake Superior, USA | 6/2013–5/2016 | $a(\lambda)$, $a_{\mathrm{cdom}}(\lambda)$, $a_{\mathrm{cdom\_dis}}(\lambda)$, $a_{\mathrm{nap\_dis}}(\lambda)$, $a_{\mathrm{nw}}(\lambda)$, $a_{\mathrm{p}}(\lambda)$, $a_{\mathrm{p\_dis}}(\lambda)$, $b_{\mathrm{b}}(\lambda)$, $b_{\mathrm{bp}}(\lambda)$, $c(\lambda)$, $c_{\mathrm{nw}}(\lambda)$, $R_{\mathrm{rs}}(\lambda)$ | HyperOCR spectral radiometers, WET Labs AC-S, WET Labs ECO-BB9, WET Labs ECO-FL3, SeaBird CTD 37SI, Perkin Elmer Lambda 35 | Mouw et al., 2017 |
| Schaeffer | Florida Estuary Optics | Northern Gulf of Mexico estuaries, USA | 9/2009–11/2011 | $a_{\mathrm{cdom}}(\lambda)$, $a_{\mathrm{nap}}(\lambda)$, $a_{\mathrm{p}}(\lambda)$, $a_{\mathrm{ph}}(\lambda)$, $R_{\mathrm{rs}}(\lambda)$ | HyperSAS, HyperPRO in profiling mode, Shimadzu UV1700 | Astuti et al., 2018; Conmy et al., 2017; Keith et al., 2014; Keith et al., 2016; Le et al., 2015; Le et al., 2016; Mishra et al., 2014; Schaeffer et al., 2015 |

[revised manuscript text omitted]